# Saporin from *Saponaria officinalis* as a Tool for Experimental Research, Modeling, and Therapy in Neuroscience

**DOI:** 10.3390/toxins12090546

**Published:** 2020-08-25

**Authors:** Alexey P. Bolshakov, Mikhail Yu. Stepanichev, Yulia V. Dobryakova, Yulia S. Spivak, Vladimir A. Markevich

**Affiliations:** 1Laboratory of Molecular Neurobiology, Institute of Higher Nervous Activity and Neurophysiology, Russian Academy of Sciences, 119991 Moscow, Russia; lampo_love@mail.ru; 2Laboratory of Functional Biochemistry of the Nervous System, Institute of Higher Nervous Activity and Neurophysiology, Russian Academy of Sciences, 119991 Moscow, Russia; m_step_68@mail.ru; 3Laboratory of Neurophysiology of Learning, Institute of Higher Nervous Activity and Neurophysiology, Russian Academy of Sciences, 119991 Moscow, Russia; julkadobr@gmail.com (Y.V.D.); v.markevich@yahoo.com (V.A.M.)

**Keywords:** saporin, saporin-based toxins, orexin-saporin, 192IgG-saporin, sleep, pain, Alzheimer’s disease

## Abstract

Saporin, which is extracted from *Saponaria officinalis*, is a protein toxin that inactivates ribosomes. Saporin itself is non-selective toxin but acquires high specificity after conjugation with different ligands such as signaling peptides or antibodies to some surface proteins expressed in a chosen cell subpopulation. The saporin-based conjugated toxins were widely adopted in neuroscience as a convenient tool to induce highly selective degeneration of desired cell subpopulation. Induction of selective cell death is one of approaches used to model neurodegenerative diseases, study functions of certain cell subpopulations in the brain, and therapy. Here, we review studies where saporin-based conjugates were used to analyze cell mechanisms of sleep, general anesthesia, epilepsy, pain, and development of Parkinson’s and Alzheimer’s diseases. Limitations and future perspectives of use of saporin-based toxins in neuroscience are discussed.

## 1. Introduction

Current research in neuroscience requires very fine tools to dissect delicate structure of neuronal networks and determine functional role(s) of certain cell subpopulations. One of approaches that is used for solution of this task is creation of transgenic animals that either lack some gene or overexpress additional markers or proteins such as channelrhodopsins or recombinases in specific cell subpopulation. However, generation of transgenic animals is expensive and time-consuming approach which sometimes gives undesired results. For example, insertion of Cre recombinase next to nestin gene in mice leads to serious cognitive and hormonal disorders [1,2]. Another difficulty with animals that selectively express channelrhodopsin in a certain neuronal subpopulation is that “selective” activation of axons in the desired brain structure may be not so selective as it was initially thought. It was found that it may affect other brain areas due to induction of additional effects by action potentials that propagate along entire axonal tree outside of stimulation area [3]. Finally, analysis of functions of certain cell subpopulations may be performed using the transgenic animals in combination with injection of recombinant viruses, which, however, should be created specifically for every task and may be expensive and time-consuming.

Needless to say, current research in neuroscience also uses conservative approaches such as administration of selective agonists/antagonists of the studied signaling system and induction of selective degeneration of certain cell types. Conceptually, the latter approach has two complementary directions: first, creation of models of neurodegenerative diseases in which administration of some toxin leads to changes that are close in some aspect to human disease; second, analysis of consequences of cell loss to understand the role played by the cell subpopulation that was eliminated by the toxin. Ibotenic acid found in mushrooms and kainic acid extracted from seaweed are, probably, the most popular toxins that were used in neuroscience to induce lesions in various brain areas. Both of them induce excitotoxic damage of nervous tissue due to hyperactivation of glutamate ionotropic receptors; since glutamate receptors present on virtually all cells in the brain, both toxins are absolutely non-selective and selectivity is achieved only by specific injection of toxin in the desired area.

Among the toxins that have specific activity against some cell population, probably, the best known toxin is 1-methyl-4-phenyl-1,2,3,6-tetrahydropyridine (MPTP). In fact, MPTP is not itself toxic but is converted into toxin 1-methyl-4-phenylpyridinium (MPP+) by monoamine oxidase B; MPP+ is taken up by dopaminergic neurons via dopamine transporter, where it interferes with functioning of complex I of mitochondrial electron transport chain leading to disruption of cell metabolism and cell death. However, MPTP, in a sense, is a unique example of selective neurotoxin which induces death of specific neuronal subpopulation when administered systemically [4]. The majority of toxins that induce cell death are not so selective and have to be modified in some fashion to gain selectivity. 

A large group of modified toxins that can cause selective degeneration of certain cell subpopulations includes conjugates of ribosome-inactivating proteins with antibodies or peptides. In general, search for plant toxins that can induce cell death via arrest of protein synthesis was, to a great extent, inspired by hope to find cure for cancer among these toxins. This direction of studies is actively developed, however, despite several clinical trials this approach is not widely used in cancer treatment (for reviews, see [5,6,7]). Of course, like other experimental tools, saporin-based toxins, which will be discussed in this review, are not flawless and here we will try to summarize advances in neuroscience made using these toxins as well as limitations of this tool.

## 2. Short Characteristics of Ribotoxins

Ribosome-inactivating protein toxins (ribotoxins) are subdivided into two types, type I and type II. Type I toxins have only A-chain which has N-glycosidase activity, whereas type II proteins have both A-chain with enzymatic activity and B-chain, which gives the protein ability to enter cells [8]. There is a heterogeneous group of type III ribotoxins, which includes ribosomal toxins with A-chain and additional chain with structure differing from B-chain [9]. Ribotoxins have very common mechanism of action: they specifically remove the A4324 adenine residue, which forms part of a tetranucleotide G(A4324)GA sequence in the 28S rRNA in the 60S subunit of the rat ribosome, a sequence that is universally conserved among eukaryotic rRNA [5]. Since type II ribotoxins can target many types of cells and it is technically challenging to eliminate B-chain from purified protein to produce cell-impermeable toxin, so far, they were not widely used in neuroscience. In contrast, saporin, which is related to type I ribotoxin, is cell impermeable and, therefore, may be targeted to the desired cell population by conjugation with various proteins such as antibodies or peptide ligands.

Saporin, which is extracted from seeds of plant *Saponaria officinalis* [10], has become one of the most popular type I ribotoxins for generation of various conjugates that can induce selective death of chosen cell population. *Saponaria officinalis* express several variants of saporin and only one, saporin-6, was selected for generation of toxin [10] because it appeared to be stable and resistant to various modifications compared to other type I ribotoxins [11]. After saporin transportation to the cytoplasm, it inactivates ribosomes by introducing break in the rRNA and, possibly, fragmentation of DNA [12], which leads to arrest of protein synthesis in the target cell and, finally, cell death. It was also shown, however, that saporin-induced cell death may be not related to translation arrest but rather to direct induction of apoptosis [13].

The idea that saporin-protein/peptide conjugate may be used to induce death of selected cell subpopulation was fruitfully developed in neuroscience for development of models of various neurodegenerative diseases and analysis of functions of various cell subpopulations. The list of conjugates used to induce death of various neuronal subpopulations is shown in Table 1 and Table 2. Generally, there are two approaches for creation of conjugated toxins: conjugation of foreign monoclonal antibodies (immunoglobulin G (IgG) from other species) with saporin or conjugation of some native peptide ligand with saporin. The first variant of conjugated toxins is called immunotoxins. Both variants of toxins are based on the idea that ligand (either IgG or peptide) interacts with target protein on the cell surface, then the toxin-surface protein complex is internalized in the cell where saporin inactivates ribosomes, which finally leads to cell death. The common approaches used to create saporin-based toxins include formation of disulfide bonds between the ligand(s) and saporin [14,15,16,17] or avidin-biotin system (biotinylated saporin + streptavidin-coated ligand) [18].

## 3. Peptide-Saporin Conjugates

The elegant idea to use natural peptide ligands to target toxins to certain cell subpopulations appeared to be very fruitful and various toxins were created on the basis of this concept. However, it should be mentioned that the majority of designed peptide-saporin conjugates are not specific (see Table 1, for list of cell populations targeted by each toxin). The lack of specificity of peptide-saporin conjugates results from wide expression of receptors of peptides used for targeting of toxins. In fact, it is really hard to expect that some set of receptors will be highly selectively expressed in one cell subpopulation in the entire brain which, of course, affects the use of peptide-saporin conjugates in the CNS. In all studies, the problem of selectivity of toxin action was resolved by selective administration of the toxin in the desired CNS structure. 

The only described problem related to the use of peptide-saporin conjugates is the loss of toxin selectivity due to cleavage of peptide used to create toxin. This problem was described for substance P-saporin conjugate and resolved by creation of cleavage resistant substance P peptide and synthesis of cleavage-resistant saporin conjugate on its basis [34]. 

We mentioned above that the majority of peptide-based saporin conjugates are not selective; however, it has to be also stressed that not all peptide-saporin toxins are absolutely non-selective. First of all, practically all peptide-saporin conjugates were created to attack neurons but not other cells present in the CNS such as astrocytes, microglia, oligodendrocytes, vascular cells, etc. Second, practically in all studies, saporin-conjugated toxin selectively eliminated some neuronal subpopulation within studied CNS structure and lack of selectivity only means that neurons in other brain structures may also be affected by this toxin if it is injected there. Importantly, single cell RNA-seq studies of the neocortex, hippocampus, other brain areas [35,36,37,38,39] and spinal cord [40] as well as RNA-seq studies of microdissected tissue pieces [41] may help to identify other cell subpopulations in the brain that may be attacked by already existing toxins or create new toxins that will be selective for some cell subpopulation. For example, oxytocin-saporin was used to eliminate neurons that receive projections from hypothalamic paraventricular nucleus in the brainstem [23]. However, this toxin may also be used to study functions of other neuronal subpopulations because oxytocin receptors are also present in the majority of brain structures and their expression in the neocortex and hippocampus is restricted to selective subpopulations of neurons, such as somatostatin-positive interneurons [37], some pyramidal neurons in the neocortex and hippocampus [36,37,41], and mossy cells in the dentate gyrus [41]. Similarly, galanin-saporin, a conjugate of hypothalamic peptide with saporin [24], may have selective action in the mature brain since its receptors are predominantly present on the somatostatin-interneurons in the neocortex and hippocampus [38] and neurons in the globus pallidus and thalamus [39]. Presumably, future analysis of oxytocin- or galanin-related functions of mentioned neuronal subpopulations may be performed using these neurotoxins. 

## 4. IgG-Saporin Conjugates

The second group of saporin-containing toxins is conjugates of saporin with immunoglobulin G called immunotoxins. This group is very heterogeneous in terms of cell subpopulations that are eliminated by them as well as in terms of molecular targets on the cell surface that are used to attract immunotoxin (Table 2). The immunotoxins have a subgroup that is attracted by cell surface receptors, like the aforementioned peptide-saporin conjugates, however, instead of peptide ligand, they contain IgG that binds with cell receptors such as NGFR/p75 [14,28,29] or integrin-alpha_M_ [33]. Another subgroup includes IgG against vesicular proteins such as dopamine β-hydroxylase (DβH) [31] and vesicular GABA transporter (VGAT) [16]. The last subgroup of immunotoxins interacts with other surface proteins such as neuronal glycoprotein Thy-1 [27] or transporters of neurotransmitters such as dopamine transporter [30] and GABA transporter GAT1 [32]. The majority of these immunotoxins was generated using specific monoclonal antibody against desired antigen; however, it was also shown that it is possible to create immunotoxin using a mixture of antibodies against one antigen. The only immunotoxin created so far using this approach is saporin-conjugated anti-VGAT-C antibodies (SAVA) which appeared to be effective and induce selective loss of GABAergic cells without affecting other neuronal subpopulations [16].

The differences between peptide-based and IgG-based saporin conjugates are not only structural but also related to basic principle of choice of cells that will be eliminated. The peptide-based toxins attack cell subpopulations with certain receptors, i.e., the targeted cell subpopulation is eliminated because cells in this subpopulation are able to respond to some particular molecular signals. As for IgG-based toxins, they remove cells on the basis of the presence of some surface molecular markers that can be either receptors or other surface molecules such as transporters. This means that peptide-saporin conjugates are an effective tool for removal of cells that receive some signals whereas immunotoxins are more versatile and may be used to remove cells that have specific non-receptor surface antigens. From neuroscience viewpoint, this means that peptide-saporin conjugates may be preferentially used to selectively eliminate neurons that receive specific, for example peptidergic, input from other neurons, whereas IgG-saporin conjugates are more preferable for elimination of cells that have some specific function, like GABA or dopamine synthesis, or specific epitope. 

We also would like to stress an important aspect of the use of saporin-based toxins compared with other lesioning agents (or with the transgenic mice approach)—the use of saporin-based conjugates offers the advantage of producing graded lesions, thus making possible to address compensatory responses by surviving neuronal populations following partial lesions as well as the effects of cell or neurotransmitter replacement strategies following complete ablations [42]. It is also worth to note that animal models with graded lesions may be used to create tests for diagnostics of pathologies at early stages [43].

## 5. Saporin Conjugates as a Tool for Experimental Research, Modeling, and Therapy in Neuroscience

### 5.1. Sleep

Sleep is a complex physiological phenomenon that involves sophisticated interaction of different neuronal subpopulations. Neuropeptide hypocretin/orexin regulates arousal and wakefulness [44] and its deficit is thought to be one of causes of narcolepsy [45]. Orexin is predominantly synthesized by neurons localized only in the lateral hypothalamus wherefrom they innervate multiple regions implicated in the regulation of sleep/arousal. However, it was not clear which neuronal subpopulations that receive hypocretin/orexin are involved in mechanisms of sleep and which functions do they perform. First of all, it was shown that removal of hypothalamic neurons that have orexin receptor OX2R with hypocretin-2-saporin induces narcolepsy-like sleeping behavior [46,47], which may become a good model for examination of drugs that correct sleeping disorders (see below). Second, analysis of hypocretin-responding neurons in the ventrolateral periaqueductal gray showed that inhibitory neurons in this area control duration of rapid-eye movement sleep and their removal does not induce cataplexy [48]. Third, elimination of neurons that have orexin receptors in the *substantia nigra* leads to hyperlocomotion and insomnia [49]. In contrast, removal of OX2R-positive neurons in the medial septal area, which are predominantly parvalbumin-positive GABAergic neurons, did not affect sleeping behavior [21]. Finally, lesion of OX2R-positive neurons in the ventrolateral preoptic nucleus (VLPO) induced insomnia [50] supporting previous data on the important role of this nucleus in sleep [51]. All these data are summarized in Figure 1. As for other signaling systems, it was shown that simultaneous elimination of neurons in three brain areas, which were previously thought to be critically involved in maintenance of arousal (basal forebrain cholinergic neurons, noradrenergic neurons in the locus coeruleus, and histaminergic neurons in the tuberomammillary nucleus), had practically no effect on the sleeping behavior [52]. Thus, different saporin-based conjugates helped to reveal neuronal subpopulations critically involved in sleep. 

An important aspect of sleep research is related to analysis of adenosine accumulation in the basal forebrain, which is thought to be one of important regulators of sleep. It was shown that removal of OX2R-positive neurons in the lateral hypothalamus prevented adenosine accumulation in the basal forebrain and increased sleep suggesting that adenosine accumulation in the basal forebrain is not critical for sleep, at least under studied conditions [53]. It was also shown that selective lesion of basal forebrain cholinergic neurons prevents adenosine accumulation in the basal forebrain and weakly alters homeostatic sleep response again supporting the idea that adenosine accumulation is not critical for sleep [54,55].

Most importantly, recent studies used the aforementioned models of narcolepsy-like behavior, which are induced by orexin-saporin injections in the hypothalamus, to find possible drug treatments for sleeping disorders in humans. Cannabidiol, a phytocannabinoid which lacks strong psychic effects, was found to decrease sleepiness [56] as well as inverse agonist of histamine H3 receptors appeared to have pronounced wake-promoting effect [57] in the saporin-based model of narcolepsy-like behavior.

### 5.2. General Anesthesia

Despite intense studies, the mechanism of action of general anesthetics remains still obscured. One of important questions that arise during analysis of mechanisms of action of general anesthetics is whether sleep and general anesthesia have common mechanisms. Trying to resolve this question and understand mechanisms of general anesthesia, researchers also used saporin-based conjugates. It was shown that orexin-saporin-induced lesion of OX2R-positive neurons in VLPO sensitizes animals to the general anesthetic effects of isoflurane [50]. The results of this study also have another important aspect, since, as we mentioned above, VLPO neurons play critical role in sleep and their removal causes insomnia, however, does not prevent general anesthetic effect of isoflurane. The latter clearly means that general anesthesia induced by inhalation of isoflurane have mechanisms that substantially differ from sleep. Similar results were obtained after injection of galanin-saporin in VLPO, which also caused death of majority of VLPO neurons [58]. It was shown that, at the initial stages, death of VLPO neurons increases resistance to isoflurane anesthesia but, in 24 days, probably, due to accumulated sleep debt, it led to increased sensitivity to isoflurane [58]. The sensitivity to isoflurane anesthesia but not to other anesthetics (propofol, pentoparbital, and ketamine) may also be increased by lesion of OX2R-positive histaminergic neurons in the tuberomamillary nucleus [59] suggesting that different general anesthetics may act via different signaling systems.

Sedative potency of propofol decreased after elimination of basal forebrain neurons using 192IgG-saporin suggesting that anesthetic effect of propofol is partly dependent on the activity of cholinergic system [60]. However, in the following study, the authors made the opposite findings—192IgG-saporin-induced lesion of cholinergic neurons in medial septum and diagonal band of Broca (MS/DBB), which predominantly innervate the hippocampus, potentiated anesthetic effect of propofol [61]. The authors related the contradiction observed to the differences in the concentration of propofol used in their studies: potentiation of propofol anesthesia was observed only at anesthetic but not subanesthetic propofol doses. Later on, it was also shown that anesthesia induced by propofol is significantly potentiated at subanesthetic dose in animals with lesioned cholinergic neurons in the nucleus basalis of Meynert (NBM) innervating neocortex [62]. Probably, mechanisms of action of an anesthetic might differ from one brain area to another, which may result in a dose-dependent appearance of behavioral effects elicited by the anesthetic and different dependence on the other signaling systems involved in its action. 

### 5.3. Epilepsy

Epilepsy is a chronic non-infectious brain disease. It remains one of the most widespread neurological diseases with about 50 million patients around the world [63]. Epilepsy is expressed as repeated partial or generalized seizure attacks; the latter are frequently associated with loss of consciousness and autonomic control. Epilepsy may be a consequence of traumatic brain injury, genetic predisposition, developmental malformations, brain infections, and tumors. One of unsolved problems that hampers the development of epilepsy treatment is the absence of relevant models of epilepsy. Recently, an attempt was made to create an experimental model of epilepsy using a saporin-based toxin. This attempt is based on the suggestion that impairments in specific neurotransmitter systems can also be followed by epileptogenesis. Thus, removal of subpopulation of GABAergic neurons in the hippocampus using stable (S)SP-saporin [34] was associated with appearance of episodes of immobilization followed by abnormally prolonged flurries of “wet-dog” shakes [64]. Brief seizures characterized by forepaw clonus and rearing, which also occurred intermittently in these SSP-saporin-injected rats, never led to occurrence of status epilepticus. Loss of GABAergic neurons was expressed, as expected, in a decrease in parvalbumin or NK1 (SSP-receptor) immunoreactivity. The authors also reported that, after a half of a year delay, convulsive behavior was associated with epileptiform discharges in the dentate gyrus and “sclerosis”-like structural abnormalities. However, this hippocampal sclerosis significantly differed from that observed in human brain samples [65]. Thus, it cannot be excluded that multiple application of even low doses of saporin-containing conjugate may evoke more progressive structural changes which are not directly related to loss of GABAergic parvalbumin-containing basket cells.

Interestingly, some types of seizures could be prevented using selective disruption specific neuronal subpopulations. We mentioned above (Table 1) that corticotropin-releasing hormone (CRH)-saporin conjugates have low selectivity because of the presence of CRH receptors in various neurons in the brain. However, injection of CRH-saporin into the cerebral ventricles resulted in a decrease in the number of CRH-sensitive cells in the lateral septum. Loss of these neurons was followed by attenuation of seizures in E1 mice, a model of idiopathic simple reflex epilepsy which arises from the interaction of genetic predisposition and lifetime exposure to stressful seizure triggers such as tail suspension handling [66]. Similar effect was also observed after intraamygdalar injection of CRH-saporin in the same strain of mice [67]. Taken together, these results support strongly the possibility that activated brain stress neuropeptide systems are necessary for the expression of neurological perturbations in seizure susceptible El mice.

### 5.4. Pain

Studies on the mechanism of pain and its treatment are another illustration of an important role of saporin conjugates application in neuroscience. Neuropathic pain is caused by a lesion or disease of the somatosensory system, including peripheral fibers (Aβ, Aδ and C fibers) and central neurons, and affects 7–10% of the general human population [68]. Multiple causes of neuropathic pain have been described; however, only multidisciplinary approach to the management of neuropathic pain can provide progress in the understanding of the pathophysiology of neuropathic pain and the development of new diagnostic procedures and personalized interventions. The situation is also complicated by the fact that at least two different sensory neuronal populations are thought to be responsible for transduction of noxious stimuli (Figure 2A). Both subpopulations project to the lamina I/II in the spinal cord but have different characteristics: the first subpopulation releases substance P (SP) and calcitonin-gene related peptide (CGRP) whereas the second subpopulation was identified by binding of isolectin IB4 (IB4) and dependence on GDNF. Spinal neurons expressing the NK1 receptor are one of key cell subpopulations involved in SP-mediated pain sensitivity. It was shown that infusion of SP-saporin into spinal cord eliminated these neurons and attenuated responses to highly noxious stimuli and mechanical and thermal hyperalgesia (Figure 2C) [19]. Later on, these results were extended by other groups using other animal models. Intrathecal administration of SP-saporin to monoarthritic rats inhibited secondary hyperalgesia, accompanied by an increase in nociceptive activation of spinal neurons expressing NK1 receptors, probably due to their destruction [69]. Neurons in the superficial dorsal horn that express NK1 receptors participate in the development of behavioral hypersensitivity following peripheral sensitization of nociceptors. Injection of a modest dose of SP-saporin into the lumbar subarachnoid space produced a partial loss of lamina I/II NK1 receptor-expressing dorsal horn neurons, attenuated thermal nociceptive sensitivity, and prevented secondary hyperalgesia when studied with an operant algesia assay [70]. Later on, it was shown that intrathecal administration of SP-saporin to dogs with bone cancer pain produces a time dependent anti-nociceptive effect with no evidence of development of deafferentation pain syndrome [71]. 

A complementary approach to decrease sensitivity to noxious stimuli, in addition to elimination of SP-sensitive neurons in the spinal cord, is to remove IB4-binding sensory neurons in the dorsal root ganglion. It was shown that elimination of these neurons by injection of IB4-saporin decreased sensitivity to noxious stimuli [72] and duration of mechanical hyperalgesia [73], however, quite unexpectedly this suppression of pain response was compensated at later stages and nociceptive threshold recovered [72]. Later on, it was shown that this compensation of IB4-saporin-induced loss of nociceptive fibers includes an increase in the surface expression of NK1 receptors in the subpopulation of spinal cord neurons that already expressed NK1 receptors (Figure 2B) [74]. Importantly, using this saporin-based toxin it was shown that GDNF is one of endogenous mediators that can activate IB4-binding nociceptors [75]. Finally, a recent study, which also utilized IB4-saporin, showed that IB4-positive nerve fibers are not involved in normal mechanical nociception but play a pivotal role in mediating mechanical inflammatory hypersensitivity [76]; however, the results of this study may have additional interpretation due to non-neuronal effects of IB4-saporin (see below).

An important problem related to pain is development of drugs that suppress pain; one of these drugs is opioids, however, it is known that administration of opioids have severe adverse effects including hyperalgesia. The neurobiological basis of opioid-induced hyperalgesia was also studied using saporin-conjugates. It was shown that systemic fentanyl, a powerful µ-opioid receptor agonist, induces hyperalgesic priming due to long-lasting modification of nociceptor function. Destruction of cells expressing peptidergic NK1 receptors or non-peptidergic nociceptors using [Sar^9^, Met(O_2_)^11^] stable (S)SP-saporin and IB4-saporin, respectively, prevented central terminal priming evoked by intrathecal fentanyl treatment [77]. However, in addition to non-peptidergic nociceptors, IB4 recognizes α-D-galactosyl residues on the surface of non-neuronal cells, such as microglia [78]. Microglia of the dorsal horns of the spinal cord is involved in the modulation of the neuropathic pain syndrome [79]. Thus, some effects of IB4-saporin in this model may be due to its anti-microglial action (Figure 2B). This conclusion is supported by finding that removal of spinal cord microglia using Mac1-saporin conjugate in combination with blockage of BDNF-TrkB signaling can reverse pain hypersensitivity after nerve injury [80].

These and other studies (for review, see [81]) laid the basis for the idea that the intrathecal injection of SP-saporin to terminal cancer patients may be used to remove NK1-positive neurons in the spinal cord to alleviate intractable pain. On the one hand, this treatment does not remove causes of pain but, on the other hand, it may considerably improve quality of life of patients and may be considered as a variant of palliative help. Phase 1 clinical trials stopped in 2016 in the United States, however, so far no results were published, probably, due to analysis of *postmortem* samples.

Analysis of other signaling systems involved in pain and itch was also performed using other saporin-based conjugates. Bombesin-saporin (BB-SAP) specifically eliminates gastrin-releasing peptide receptor (GRPR) expressing neurons [82]. In the spinal cord, these neurons are critically involved in both histaminergic and nonhistaminergic sensation of itch [83]. After intrathecal injection to ovalbumin-sensitized mice, BB-saporin or SP-saporin ablated GRPR- or NK1R-expressing spinal neurons, respectively. This, respectively, diminished the expression of hyperkinesis or chronic itch, including spontaneous scratching.

The aforementioned CRH-saporin was also used in the study that analyzed mechanisms of pain sensation. It was shown that elimination of some subpopulation of spinal cord cells by intrathecal injection of CRH-saporin attenuated spinal nerve ligation- and carrageenan-induced tactile hypersensitivity in rats [84]. Further analysis using selective antagonists of CRHR1 receptors showed that pain is mediated by interaction of CRH with CRHR1 receptors. This study supported the idea that pain is stress-inducing condition and its mechanisms involve stress-related signaling cascades even at the level of spinal cord. 

### 5.5. Anxiety and Autism Spectrum Disorders

Autism spectrum disorders are neurodevelopmental disorders that begin during childhood with long-term clinical and social implications for affected individuals, their families, and the community. Presently 1 in 160 children has an autism spectrum disorder [85]. Psychiatric comorbidities, including anxiety and mood disorders, are also common in autism spectrum disorders [86]. However, mechanisms of these disorders remain poorly understood, probably, because of the absence of suitable animal models. Recently, the effects of progressive disruption of neuronal inhibition within the basolateral nucleus of the amygdala (BLA) on conspecific social interaction in rats was used to investigate functional networks from the ventral medial prefrontal cortex to the BLA which may be responsible for abnormal social behavior [87]. The authors targeted SSP-saporin to NK1-expressing GABAergic neurons of the BLA. Disrupted BLA inhibitory tone was followed by persistent social inhibition that was not reversed by habituation paradigm of social interaction. Thus, selective lesion of GABAergic neurons in the BLA was suggested as a model of autism. Interestingly, the selective destruction of neurons in the *nucleus incertus* with CRH-saporin is followed by impaired fear conditioning [88] and also indicates the role of this nucleus in the mechanisms of anxiety [89]. Furthermore, intraamygdalar injection of CRH-saporin to E1 mice, which decreased the CRH content in the BLA, suppressed social interaction behavior [67]. It was also shown that removal of neuropeptide Y-sensing neurons in the central amygdala increased anxiety-like behavior; in contrast, elimination of this type of neurons in the hypothalamus suppressed anxiety [90]. Taken together, these data show that saporin conjugates may be used to model impairments of social behavior due to targeted disruption of specific networks as well as to study the involvement of these networks in the development of autism spectrum disorders, including comorbid anxiety conditions.

### 5.6. Parkinson’s Disease

Parkinson’s disease (PD) is a brain disorder that is associated with tremor, rigidity, bradykinesia and postural instability. As the disease progresses, people may also have mental and behavioral changes, sleep problems, depression, memory difficulties, and fatigue. The current theory argues that PD starts in the enteric nervous system, the medulla and the olfactory bulb, which controls sense of smell and later on progresses to the *substantia nigra* and cortex [91]. Most of current animal models are based on the induction of dopaminergic (DAergic) cell loss in the *substantia nigra* as a principal neuropathological mechanism. The conjugate of a monoclonal antibody to the dopamine transporter (anti-DAT) with saporin was used for microinjection into either the center of the left striatum or the left lateral ventricle of adult male rats [30]. Both types of injections produced destruction of DAergic neurons in the ipsilateral *substantia nigra* with no significant non-specific effect for other monoaminergic structures. DAergic deficit was associated with decreased locomotor activity in the open field test. Recently, an alternative saporin-based approach to induction of death of DAergic neurons in the *substantia nigra* was proposed: local injection of quantum dots conjugated with saporin. It was shown that quantum-dots-saporin injected in the *substantia nigra* induces loss of DAergic neurons and activation of microglia as well as loss of motor coordination [18].

Despite extensive studies L-DOPA remains the primary treatment for PD patients. However, therapeutic benefits of L-DOPA are compromised by the development of abnormal involuntary movements known as L-DOPA-induced dyskinesia. Several groups have shown that this might be due to additional damage to the norepinephrine (NE) system. They induced degeneration of DAergic neurons using 6-OHDA into striatum in combination with intraventricular injections of a powerful NE immunotoxin, anti-DBH-saporin, to eliminate the NE neurons in the *locus coeruleus*, and associated pontine nuclei. Animals with combined DA and NA lesions were more prone to develop L-DOPA-induced dyskinesia, even at low L-DOPA doses, and they performed significantly worse in tests of reflexive and skilled paw use, the stepping and staircase tests, compared to DA-only lesioned rats [92,93]. These findings suggest that severe NE loss may reduce L-DOPA treatment efficacy and demonstrate that degradation of the NE system is an important consideration when evaluating L-DOPA effects in later stage of PD.

Additional important feature of 6-OHDA-induced PD animal model is a reduction in the neuronal profile within the brainstem ventral respiratory column with a decrease in the hypercapnic ventilatory response. Orexin-expressing cells of the lateral hypothalamus/perifornical area (LH/PeF) are involved in the regulation of breathing in these animals. Injection of orexin-saporin conjugate into the LH/PeF resulted in degeneration of these neurons. It allowed to reveal an important chemoreceptor function of hypothalamic orexin neurons in the control of breathing response to hypercapnia during the dark (active) phase of a day cycle [94]. Taken together, these data show that application of saporin conjugates allowed to reveal some delicate details of pathogenesis of PD.

### 5.7. Alzheimer’s Disease

Alzheimer disease (AD) is the most common form of dementia and may contribute to 60–70% of dementia cases. According to WHO data, around 50 million people worldwide have dementia, and there are nearly 10 million new cases every year [95]. Loss of cholinergic neurons in the basal forebrain (BFCN) is one of the most important pathogenetic features of AD. AD is a complex disease that is characterized by a substantial perturbation of neurotransmitter metabolism. Loss of cholinergic neurons in the basal forebrain, primarily in the nucleus basalis of Meynert, is one of the most important pathogenetic features of AD. However, neurodegeneration is also observed in the entorhinal cortex, hippocampus, and *locus coeruleus*. Mounting evidence indicates that the loss of noradrenergic innervation greatly exacerbates AD pathogenesis and progression, although the precise roles of noradrenergic components in AD pathogenesis remain unclear [96,97,98].

Modeling of cholinergic dysfunction allows to reproduce memory impairments, which are among the central symptoms of AD associated dementia. Several neurotoxic substances were used in order to produce loss of BFCN in animals [99]. Till the mid of 1990′s, ethylcholine aziridinium ion (AF64A) was widely used for this purpose, the toxin with a structure similar to that of choline. Due to this similarity AF64A is transported into the cell by the system of high-affinity choline uptake [100]. Intracerebroventricular administration of AF64A results in cholinergic deficit and disturbances of learning and different forms of memory. Inside the cell AF64A exerts its neurotoxic action via alkylation of proteins and DNA, and induction of oxidative stress. Thus, AF64A not only has amnestic action, but allows modeling some neuropathological mechanisms, which are reminiscent of those in AD. However, specificity of AF64A cholinotoxicity is not very good and a relatively high concentration of AF64A may result in death of other populations of neurons such as monoaminergic cells. Starting from 1990′s, a new generation of neurotoxins became very popular, specifically those that are based on immunoconjugates of saporin. Presently, saporin conjugates are among the tools with the most selective mode of action.

The main idea underlying selective toxicity is again conjugation of saporin with the antibody recognizing a pan-neurotrophin p75/NGFR receptor. This receptor is predominately expressed by cholinergic neurons in the adult mammalian brain [35,101,102]. There are three agents that selectively kill NGFR-positive cholinergic neurons: 192IgG-saporin for rats [14], mu-saporin for mice [28], and ME20.4-saporin for primates [29]. 

One of them, mu-saporin, induces more than 80% loss of choline acetyltransferase (ChAT)-positive neurons in the medial septum and more than 50% in the nucleus basalis in mice. Preserved parvalbumin immunostaining suggests that the lesion is specific to BFCNs [103]. The mu-saporin-treated mice display significantly impaired spatial learning/memory performances in a water maze and in a Barnes maze with slowed down learning and altered memory retention. A loss of BFCNs is accompanied by simultaneous activation of microglia and astrocytes in the basal forebrain region and minocycline, a second-generation tetracycline with known anti-inflammatory and neuroprotective properties, attenuates mu-saporin-induced neurons loss, glial activation, and transcription of downstream pro-inflammatory mediators. In addition to neuroprotection, minocycline treatment mitigated the cognitive impairment that appears to be a functional consequence of mu-saporin lesioning [104]. 

Another saporin conjugate is 192IgG-saporin which is used to selectively kill cholinergic neurons in rats. Like mu-saporin, this is an efficient and selective immunotoxin for the NGF-receptor bearing cholinergic neurons in rat basal forebrain. Intracerebroventricular (ICV) administration of the 192IgG-saporin conjugate appears to induce a nearly complete and specific lesion of neocortical and hippocampal cholinergic afferents. Other neuronal systems in the basal forebrain are spared by the immunotoxin [105]. Starting from the early 1990-s the behavioral effects of this neurotoxin have been studied in a variety of learning paradigms in several dozens of experiments. Injection of immunotoxin exerts little or even no effect on working memory performance in rodents [106,107,108,109,110]. Cholinergic lesions in the MS/DBB impair acquisition of the delayed matching to position task in both males and females [110,111,112]. In addition, 192IgG-saporin induces motor deficits, hyperactivity and reduced T-maze alternation [113,114] as well as impairs performance in the Morris water maze task and passive avoidance retention [115,116]. 

Administration of 192IgG-saporin into the cerebral ventricles or NBM/MS results in a different degree of cholinergic denervation in the neocortex or hippocampus. Pizzo et al. have demonstrated a dynamic interplay between the severity of cholinergic deficit and task demands revealing different types of mnemonic impairments. Water maze performance was similarly impaired for ICV- and intra-NBM/MS animals during various phases of testing, whereas animals with individual lesions of the NBM or MS performed at the level of controls [117]. Despite this mostly allocentric type of learning, the egocentric-based T-maze task revealed a significant group difference between the ICV- and the intra-NBM/MS animals. The later animals showed a severe deficit in the non-match- and match-to-position version, whereas again, animals with single lesions were unimpaired. This conclusion was additionally supported by other authors who injected 192IgG-saporin into the NBM [110]. Presumably, cognitive impairments observed in this model of AD might be due to the delayed effects of immunolesion on hippocampal functions. 192IgG-saporin administration significantly inhibits neurogenesis in the olfactory bulbs [118] and hippocampus [119], which was prevented by neurotrophin infusion [118].

Application of mu-saporin allowed to significantly improve an AD transgenic mouse model that overexpresses mutant amyloid precursor protein (APP) and presenilin-1 with deleted exon 9 (APPswe/PS1dE9). Selective depletion of ChAT-positive neurons by mu-saporin in APPswe/PS1dE9 mice produced a more complete model of the disease because amyloid β (Aβ) deposition and tau phosphorylation were significantly increased after the lesion of cholinergic neurons [120]. Enhanced perivascular Aβ accumulation was also observed in the neocortex of rabbits in six months after lesion of NBM neurons with ME20.4-saporin [121]. These data suggest that BFCN are somehow involved in the regulation of Aβ level; one of probabilities is that BFCN fibers serve as a source of NGFR ectodomain (p75ECD), which suppresses β-secretase involved in Aβ formation [122], and their elimination releases this blockage of β-secretase. However, the mechanisms of amyloidosis may vary under different conditions and may also involve impaired functioning of blood vessels and blood-brain barrier. For example, it was recently shown that amyloidosis is enhanced in the frontal cortex of Tg344 rats (rats overexpressing APPswe/PS1dE9) after elimination of noradrenergic projections by anti-DBH-saporin [123]. The authors also found that cortical noradrenergic deficit was associated with compromised function of blood-brain barrier and remodeling of vessel structure, which resembled pathological stenosis. These data as well as our data that cholinergic deficit induced by 192IgG-saporin may lead to altered expression of vascular transport proteins [124] point to an important role that may be played by disturbed vascular function in AD pathogenesis.

Cholinergic denervation of the cerebral cortex due to degeneration of BFCNs evoked some compensatory changes in the expression of various neurotransmitter receptors, including the increased expression of M1 and M2 muscarinic AChRs, AMPA, kainate, and GABAa receptors whereas the expression of NMDA receptors decreased. These changes were more prominent in the cortical regions displaying a reduced activity of AChE and decreased levels of high-affinity choline uptake sites due to immunolesion [105]. These findings are contrasted by latter studies where changes in mRNA levels were studied using either microarrays or RNA-seq. It was found that lesion of BFCN was not associated with changes in mRNA level of the mentioned channels in the frontal cortex [125]. Similarly, MS/DBB and MS/NBM lesions also did not alter mRNA expression of majority of mentioned channels in the hippocampus [126,127]. Presumably, loss of cholinergic afferentation altered translation of some mRNAs but this hypothesis requires further examination.

ACh plays an important role in the regulation of inflammation in peripheral tissues. It has been suggested that ACh is also involved in the regulation of glia functions in the CNS. Therefore, selective lesion of BFCNs by 192IgG-saporin may be a suitable tool to discover a role of glia in the AD pathogenesis. In the previous studies, a significant response of microglia in the site of immunotoxin injections was detected using immunohistochemistry and lectin histochemistry [128]. Microglial cells in the lesion exhibited phagocytic activity supported by specific immunohistochemistry [129] and electron microscopy [130], but this effect significantly depended on the delay after the toxin injection. However, neither microglial nor astroglial cells at the lesion site expressed interleukin (IL)-6 mRNA or leukemia inhibitory factor mRNA [131] studied using combined in situ hybridization and immunochemistry. Additionally, neither IL-1β nor IL-6 expression could be detected in any of the activated glial cell types in the hippocampus following immunotoxic cholinergic lesion, whereas IL-1α was found to be expressed in astroglial cells only [132]. 

Recently, we have applied a transcriptomic approach to study whether 192IgG-saporin induces neuroinflammatory consequences in the hippocampus. Administration of immunotoxin into the cerebral ventricles resulted in moderate impairments of cognitive functions which were associated not only with loss of cholinergic neurons in the medial septum, but also with loss of neurons in the CA3 field of the hippocampus [124,126]. Microglia proliferation was observed in the dorsal hippocampus and in the white matter. Microglia activation was related to the expression of a subset of genes associated with inflammation specifically in dorsal hippocampus; however, it did not include the genes encoding mediators of acute inflammation, such as IL-1β, IL-6, IL-15, IL-18 or TNFα [126]. Data of transcriptomic analysis were further supported by the results of real-time PCR [124]. Microglia activation could also be detected in the neocortex [133]. Taken together these data indicate that loss of cholinergic projections to the hippocampus leads to proliferation of microglia cells with phagocytic activity in the hippocampus; however, these cells do not express typical proinflammatory signaling molecules. It is possible that the absence of typical inflammatory phenotype of microglia discovered using 192IgG-saporin can help to understand why ordinary non-steroid anti-inflammatory drugs are not efficient in AD.

We already mentioned above that noradrenergic deficit is an important feature of AD. The pathological mechanisms related to noradrenergic deficit were studied using anti-DBH-saporin. Administration of the anti-DBH-saporin to neonatal or young rats was followed by working memory impairments, although the lesioned rats exhibited normal behavior in spatial reference memory tests [42,134]. This effect was associated with loss of noradrenergic innervation in the hippocampus and probably lower rate of neurogenesis. However, like in the case of 192IgG-saporin, neuroglia may strongly contribute to the impairment of memory after selective lesion of noradrenergic neurons. It has been hypothesized that degeneration of locus coeruleus neurons and insufficiency of noradrenaline supply to the hippocampus decreases functional activity of astrocytes in this brain structure and thus inhibits neurogenesis [98]. 

We already mentioned in the chapter about pain that degeneration of one subpopulation of neurons may be compensated by some changes in survived cells. It was shown that, for example, memory impairments after lesion of cholinergic NBM neurons ameliorated with time [135] suggesting that some compensation occurred. In fact, practically in all cases it is absolutely unclear how degeneration of certain cell subpopulation will be compensated. However, it was partly uncovered in the case of loss of cholinergic innervation of the hippocampus (Figure 3). It appeared that the level of nerve growth factor (NGF) protein that is synthesized and secreted by hippocampal interneurons and taken up by terminals of septal cholinergic neurons becomes elevated after degeneration of cholinergic neurons [136,137]. This elevation of NGF level in the extracellular space leads to NGF diffusion to the areas, where it, probably, diffuses very seldom under normal conditions. One of these areas is synaptic contact of sympathetic fibers, which come from the superior cervical ganglion (located outside of the brain), with blood vessels. These sympathetic fibers are noradrenergic but also express NGF receptors [138,139] and their activation by elevated NGF leads to sprouting of sympathetic fibers [140] and, probably, their transdifferentiation into cholinergic phenotype [141]. Of course, this compensation develops slowly and takes several weeks to develop but, anyway, induction of cholinergic deficit in the hippocampus finally leads to compensatory sprouting of sympathetic fibers in the hippocampus (Figure 3C). Moreover, sympathetic sprouting seem to be one of factors that help to maintain normal processing of Aβ after cholinergic degeneration [140]. Probably, this compensatory pathway is not one and only and search for other pathways may be also important in terms of understanding of mechanisms underlying AD pathogenesis.

## 6. Concluding Remarks

Here we summarized principal findings of studies that used saporin-based conjugates in different areas of neuroscience. It may be seen that some variants of toxin appeared to be very successful, like SSP-saporin or 192IgG-saporin, whereas others were used only in several studies. On the one hand, success of the toxin is related to the efficacy of the toxin binding, transport, and internalization along with the surface ligand as well as the tasks that are solved using it and applicability of the results to the development of medications or treatments. On the other hand, success is also determined by the presence of alternative tools that may be cheaper and as effective and selective as saporin-based toxin. For example, anti-DAT-saporin was not very popular because synthetic toxin MPTP appeared to be less expensive drug with similar effects.

The use of saporin-based conjugates in neuroscience has several important limitations. The first is necessity to make injections of the toxins in the brain structures, which cannot be done without cranial surgery. This limitation is determined by blockage of the toxin transport from blood into the brain by blood-brain barrier (BBB). Nevertheless, intravenous injection of immunotoxin can cause specific damage in the peripheral CNS [14]. Several approaches were proposed to increase BBB permeability for various agents: (i) focused ultrasound [142], (ii) photodynamic effect [143,144], and (iii) use of peptide aptamers [145]. However, none of these methods were used for delivery of saporin-based constructs. The problem of BBB permeability for saporin-containing toxins was addressed in recent studies [146,147] where the authors used angiopep-2-directed and redox-responsive virus-mimicking polymersomes (angiopep-2 is a peptide targeting to low-density lipoprotein receptor-related protein-1) or apolipoprotein E peptide [(LRKLRKRLL)2C], which specifically binds to low-density lipoprotein receptor members. It was shown that these two approaches can efficiently and selectively chaperone saporin to orthotopic human glioblastoma xenografts in nude mice after systemic administration. Although this allowed to inhibit efficiently glioblastoma growth, the success of those constructs was probably because a high rate of vascularization of tumor. Its specificity for targeted disruption of other cell populations was not revised. The second limitation is very slow rate of cell degeneration after administration of the saporin-based toxin; it takes several days to induce elimination of desired cell subpopulation [105,148] and this duration may depend on the route of toxin administration [55]. In this respect, saporin-based drugs are not as fast as different blockers of various receptors or transporters. The third is induction of inflammatory response in the injection area, which, however, ceases with time. The fourth is compensation that occurs after degeneration of targeted cell subpopulation. This compensatory response in the majority of cases is hardly predictable and looks like a serious limitation. However, if we look on this compensation from the other side, analysis of its mechanisms may provide important insights in the mechanisms of development of different CNS pathologies. 

In general, saporin-based toxins appeared to be a useful tool and creation of new toxins may provide alternative animal models of human diseases. For example, current variant of oligodendrocyte-targeting saporin-based toxin (cholera toxin-saporin conjugate) is not selective and kills also astrocytes and microglia [149]. It looks possible that generation of new IgG-saporin toxin against surface oligodendrocyte marker may be used to generate a variant of toxin that selectively damages oligodendrocytes to model process of demyelination in the CNS, which may help to understand some aspects of pathological process that occurs in multiple sclerosis. 

Finally, we would like to stress that recent advances in molecular neurobiology such as creation of molecular fingerprints of certain cell subpopulations in the CNS using single cell RNA-seq [35,36,37,39] may help to generate new specific toxins. It is also important to note that peptide- and IgG-based saporin conjugates may, probably, give way to more selective and effective saporin-based conjugates. Clearly, the number of small molecule ligands grows every day and some of them may be used to create saporin-conjugated toxin. The only limitation is that the selected ligand have to be endocytosed after binding with its target. Possibility of use of this sort of conjugates was recently shown [150], however, so far this approach remains at the stage of idea in neuroscience studies.

## Figures and Tables

**Figure 1 toxins-12-00546-f001:**
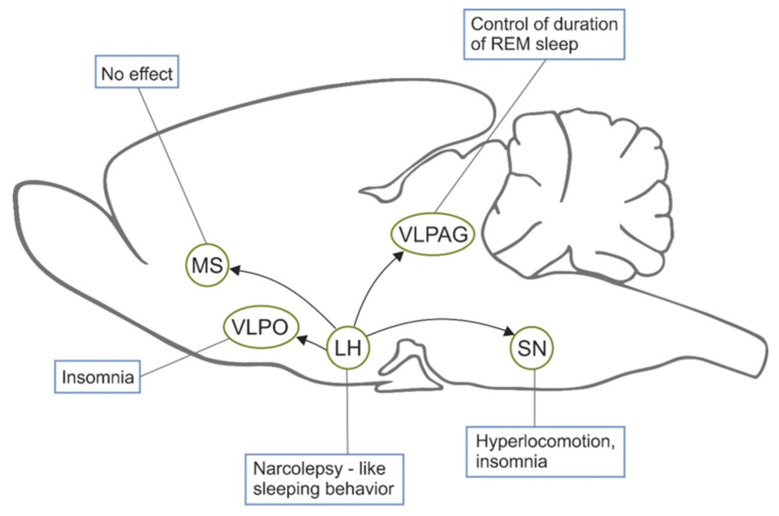
The sleep-related effects of orexin-saporin injection in various parts of the brain. The effects caused by death of neurons that receive orexinergic inputs from the lateral hypothalamus (LH) are shown next to the structure that was destroyed by the toxin. MS, medial septum; VLPO, ventrolateral preoptic area; SN, *substantia nigra*; VLPAG, Ventrolateral Periaquaductal Gray.

**Figure 2 toxins-12-00546-f002:**
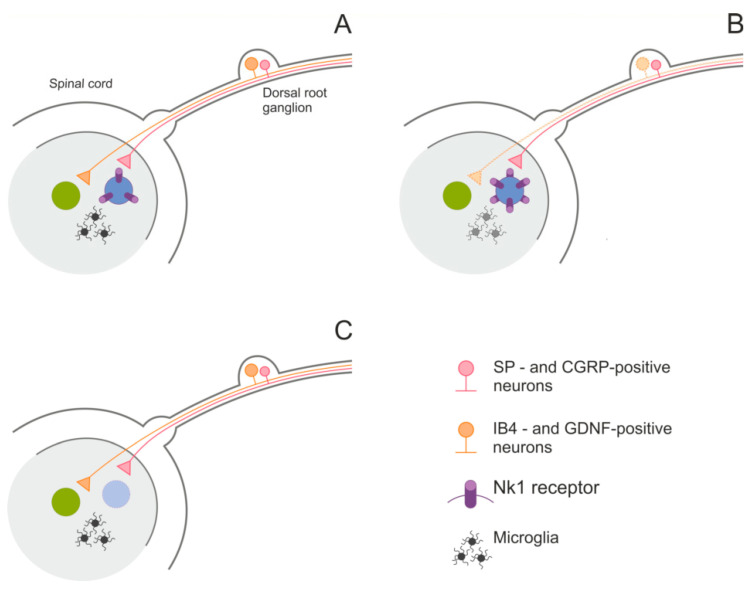
Effects of different saporin-based conjugates on the nociceptive system in the spinal cord. Panel (**A**), basic connectivity of two major types of nociceptive neurons. Panel (**B**), a compensatory increase in the expression of NK1 receptors in the spinal cord neurons after elimination of IB4-positive neurons in the dorsal root ganglion. Note that IB4-saporin may also cause degeneration of microglia in the spinal cord and affect pain sensitivity. Panel (**C**), intrathecal injection of SP-saporin causes degeneration of NK1-positive neurons in the spinal cord and reduces sensitivity to noxious stimuli.

**Figure 3 toxins-12-00546-f003:**
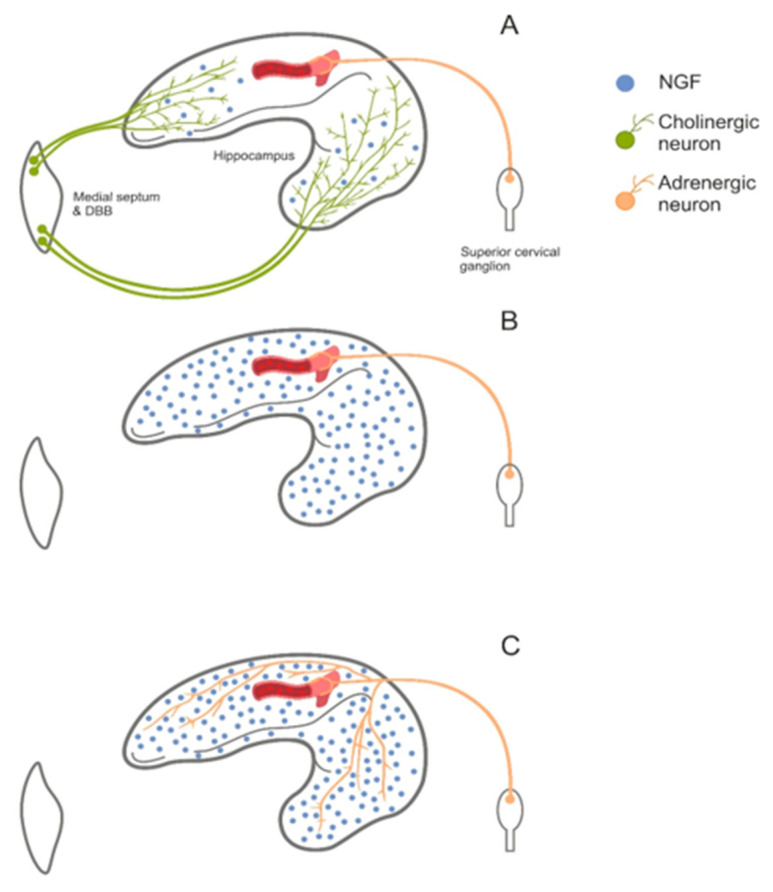
Compensation of cholinergic deficit in the hippocampus after death of cholinergic neurons in the medial septum and diagonal band of Broca (DBB). Panel (**A**), normal pattern of innervation. Panel (**B**), loss of cholinergic input leads to an increase in NGF level. Panel (**C**), compensatory sprouting of sympathetic fibers in the hippocampus resulting from elevated NGF level.

**Table 1 toxins-12-00546-t001:** List of peptide-saporin conjugates.

Saporin Conjugate(s)	Molecular Target of Toxin	Cell Subpopulation Killed by Toxin	References
Substance P-saporin	Neurokinin 1 receptor	Interneurons of the hippocampus and neocortex; spinal cord lamina I neurons	[19]
CRF-saporin	CRHR1 and CRHR2 receptors	Majority of CNS neurons	[20]
Orexin B-saporin	OX1 and OX2 receptors	Majority of CNS neurons	[21]
NPY-saporin	NPY receptors	Majority of CNS neurons	[22]
Oxytocin-saporin	Oxytocin receptors	Somatostatin-positive interneurons, some pyramidal neurons on the hippocampus and neocortex, mossy cells in the dentate gyrus	[23]
Galanin-saporin	Galanin receptors	Somatostatin-positive interneurons, neurons in globus pallidus and thalamus	[24]
Bombesin-saporin	Bombesin receptors (NMBR, GRPR, and BRS3)	GPRP-positive interneurons in the hippocampus and neocortex; neurons in the superficial dorsal horn of the spinal cord.	[25]
Cholera toxin B-saporin	GM1-ganglioside	Majority of CNS neurons	[26]

**Table 2 toxins-12-00546-t002:** List of IgG-saporin conjugates.

Saporin Conjugate(s)	Molecular Target of Toxin	Cell Subpopulation Killed by Toxin	Model of Pathology	References
OX7-saporin	Thy-1 glycoprotein	Majority of CNS neurons, Cerebellar Purkinje neurons	-	[27]
192IgG-saporin (rat) mu-saporin (mice) ME20.4-saporin (primates)	NGFR	NGFR-positive neurons; forebrain cholinergic neurons; upper cervical ganglion	Alzheimer’s disease	[14,28,29]
Anti-DAT-antibody-saporin	Dopamine transporter (DAT)	Dopaminergic neurons	Parkinson’s disease	[30]
anti-DβH-saporin	dopamine-β-hydroxylase (DβH)	Locus coeruleus/subcoeruleus	Alzheimer’s disease	[31]
anti-GAT1-saporin	GABA transporter 1 (GAT1, slc6a1)	GABAergic neurons	-	[32]
saporin-conjugated anti-VGAT-C antibodies (SAVA)	Vesicular GABA transporter (VGAT)	GABAergic neurons	-	[16]
Anti-mac1-saporin	Integrin α_M_ (Mac-1, CD11b)	Macrophages; microglia in CNS	-	[33]

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
