# Peer review of "Saporin from Saponaria officinalis as a Tool for Experimental Research, Modeling, and Therapy in Neuroscience"

_toxins, 2020, doi:10.3390/toxins12090546_

Round 1
Reviewer 1 Report
The manuscript 'Saporin from Saponaria Officinalis as a tool for experimental research, modeling and therapy in Neuroscience' is a timely update on the fascinating field of disease modeling based on selective and efficient ablation of discrete neuronal populations, whose functional role(s) can therefore be adequately investigated. The manuscript is well-written and organized, also providing a nice overview on the advantages and limitations of the use of saporin-based conjugates available today and their relevance in neurobiological research.
This Referee has just a few suggestions to the Authors, as follows:
1 - It should perhaps better stressed that the success and popularity of a toxin conjugate crucially depends on the efficacy of the binding, transport and internalization along with the surface ligand. These features, for example, have been instrumental to the wide success of 192 IgG-saporin over the years;
2 - Compared with other lesioning agents (or with the transgenic mice approach) the use of saporin-based conjugates offers the advantage of producing graded lesions, thus making possible to address compensatory responses by surviving neuronal populations following partial lesions, as well as the effects of cell or neurotransmitter replacement strategies following complete ablations. A brief discussion of such issue in the initial general part would be appropriate;
3 - In the description of immunotoxin-based modeling (particularly Alzheimer's Disease) it would perhaps be appropriate not to hold onto a single neurotransmitter (Acetylcholine) and the effects of its selective immunotoxin-induced depletion. A number of recent studies (reviewed in Gannon et al., 2015; Borodovitsyna et al., 2017: Leanza et al., 2018) have in fact addressed the increasingly important role played by the noradrenergic Locus Coeruleus in cognitive and histopathological aspects of the disease, and some studies (e.g. Coradazzi et al., 2016; Pintus et al., 2018) have started to investigate these issues using the anti-DBH-saporin toxin in experimental animals, showing, for example, important implications of the ascending noradrenergic system in working memory and the expression of pathological proteins. Thus, perhaps some more space should be given in the text to this toxin that in spite of its high cost (certainly not higher than that of other saporin conjugates!) has indeed proven far more selective and efficient than other neurotoxins and has been presented in the manuscript only as a sort of accessory tool.
Minor points:
- There seems to be some verbs or prepositions missing in various sentences (e.g. lines 42, 66, 289, 526) and some typos (e.g. 'glodus' should be 'globus' on line 124) the Authors may want to check.
Reviewer 2 Report
Manuscript ID: toxins-881693
Type of manuscript: Review
Title: Saporin from Saponaria officinalis as a Tool for Experimental Research, Modeling, and Therapy in Neuroscience
Comments to Editor and authors
In the manuscript entitled “Saporin from Saponaria officinalis as a Tool for Experimental Research, Modeling, and Therapy in Neuroscience” the authors describe very well the utilization on Saporin in neuroscience.
Major revision:
1) in the whole manuscript there is no reference to the BBB and to the possible ways to overcome it without resorting to the intracranial injection of the constructs.
2) Throughout the manuscript there is no reference to the conjugation chemistry of the various constructs with Saporin (Ab, Peptides, Ligands etc)
3)Section 2: please also describe Type III RIPs
Minor revisions:
Abstract, Lane 5: please write Saponaria officinalis in italics
Lanes 70, 72: please write Saponaria officinalis in italics
Lane 168: please write substantia nigra in italics
Figure 1, Lane 191: please write substantia nigra in italics
Lane 326: please write postmortem in italics
Lane 354: please write nucleus incertus in italics
Lanes 368,369, 376, 377: please write substantia nigra in italics
Lane 384: please write locus coeruleus in italics
Lane 477: Please delete (2002)
For these reasons, after the corrections/additions requested the manuscript can be accepted.

Reviewer 3 Report
The authors well summarize the use of Saporin and its derivatives after conjugation with specific peptides or antibodies to target selective cells populations and to be used as disease model in neuropsychiatric disorders.
The paper is well organized. There is a clear and exhaustive overview of Saporin and its use in the context of different disease models for neurodegenerative disorders. When describing the different disorders each section is introduced by a nice introduction of the conditions followed by a review of the literature relative to the use of Saporin in that context.
I would suggest only minor edits to the manuscript:
- Page 5, line 156: add “a” (“sleep is a complex”).
- Page 5, sentence line 157-158 (“Neuropeptide […] narcolepsy”): add reference
- Page 5, line 176-178: this sentence is not clear, please rephrase
- Page 7, line 231: loss of consciousness is not always present with all type of seizures, please correct
- Page 8, line 288: add ”the” (“sensory neurons in the dorsal root…”)
- Page 8, line 289: change “injection” with “injecting”
- Page 10, line 364: please change tremor with shakiness, add slowness and remove lack of coordination. The cardinal features of PD are tremor, rigidity, bradykinesia and postural instability
- Page 11, line 374: what do you mean with “low activity”?
- Page 11, line 380: change “are compromised” with “may be compromised”
- Page 14, line 524: “honestly speaking” is too colloquial, please change
- Page 15, concluding remarks: here the authors gave a critical overview also of the limitations of the use of Saporin, which are not mentioned at all in the introduction. It would be good start mentioning some of the limitation also at the beginning of the manuscript
Round 2
Reviewer 2 Report
After the additions/corrections made by the authors, the manuscript can be accepted in this form